# The *Latoia consocia* Caterpillar Induces Pain by Targeting Nociceptive Ion Channel TRPV1

**DOI:** 10.3390/toxins11120695

**Published:** 2019-11-27

**Authors:** Zhihao Yao, Peter Muiruri Kamau, Yalan Han, Jingmei Hu, Anna Luo, Lei Luo, Jie Zheng, Yuhua Tian, Ren Lai

**Affiliations:** 1Department of Pharmacology, Qingdao University School of Pharmacy, Qingdao 266000, China; 2Key Laboratory of Animal Models and Human Disease Mechanisms of Chinese Academy of Sciences/Key Laboratory of bioactive peptides of Yunnan Province, Kunming Institute of Zoology, Kunming 650223, China; 3University of Chinese Academy of Sciences, Beijing 100049, China; 4Sino-African Joint Research Center, Kunming Institute of Zoology, Chinese Academy of Sciences, Kunming 650223, China; 5College of Life Sciences, Nanjing Agricultural University, Nanjing 210095, China; 6Department of Physiology and Membrane Biology, University of California, Davis, CA 95616, USA; 7KIZ-CUHK Joint Laboratory of Bioresources and Molecular Research in Common Diseases, Kunming Institute of Zoology, Chinese Academy of Sciences, Kunming 650223, China; 8Institute for Drug Discovery and Development, Chinese Academy of Sciences, Shanghai 201203, China; 9Center for Biosafety Mega-Science, Chinese Academy of Sciences, No.44, Xiaohongshan, Wuchang District/Huangjin Industrial Park, Zhengdian Street, Jiangxia District, Wuhan 430207, China

**Keywords:** caterpillar, *Latoia consocia*, venom, trpv1, pain

## Abstract

Accidental contact with caterpillar bristles causes local symptoms such as severe pain, intense heat, edema, erythema, and pruritus. However, there is little functional evidence to indicate a potential mechanism. In this study, we analyzed the biological characteristics of the crude venom from the larval stage of *Latoia consocia* living in South-West China. Intraplantar injection of the venom into the hind paws of mice induced severe acute pain behaviors in wild type (WT) mice; the responses were much reduced in TRPV1-deficit (TRPV1 KO) mice. The TRPV1-specific inhibitor, capsazepine, significantly attenuated the pain behaviors. Furthermore, the crude venom evoked strong calcium signals in the dorsal root ganglion (DRG) neurons of WT mice but not those of TRPV1 KO mice. Among the pain-related ion channels we tested, the crude venom only activated the TRPV1 channel. To better understand the venom components, we analyzed the transcriptome of the *L. consocia* sebaceous gland region. Our study suggests that TRPV1 serves as a primary nociceptor in caterpillar-induced pain and forms the foundation for elucidating the pain-producing mechanism.

## 1. Introduction

Caterpillars represent the larval stage of Lepidopterans (moths and butterflies) that are distributed in various ecosystems around the globe [1]. Evolution of the order Lepidoptera originated from Jurassic ancestors about 200 million years ago [2]. Approximately 12 families, mainly moth caterpillars, are toxic and may lead to serious harm to human beings which include burning sensation, severe pain, edema, erythema, hemostatic disturbances, and renal failure [3]. *Latoia* caterpillars have bristles (spines) of varying sizes over their body (Figure 1A). These bristles consist of a nonporous tegument which originates from the cuticle and have a hollow canal for venom storage [4]. This differs from other insects such as wasps and bees whereby the organs that lay eggs or the stingers are attached to the venom gland and have the capacity to cause damage to people. Caterpillars lack specialized single gland cells for venom production, and instead venom is produced by secretory epithelial cells above the tegument and are stored in the bristles. The chitin-rich tips at the distal end of the bristles are easy to break, releasing the venom from the internal pipes [5]. The venomous bristles serve as a highly effective deterrent mechanism against potential predators.

Venomous animals induce pain and inflammation as part of their defense strategy against predators and competitors [6]. Ion channels are often highly sensitive targets of toxins from venomous animals. Pain-related ion channel receptors like transient receptor potential (TRP) channels, hyperpolarization-activated cyclic nucleotide-gated (HCN) channels, acid-sensing ion channels (ASICs), and voltage-gated ion channels (VGICs) are expressed in sensory neurons [7,8,9,10]. Among these channels, the transient receptor potential vanilloid 1 (TRPV1) mediates nociceptive inputs form the peripheral nervous system to the central nervous system, and serves to integrate diverse painful stimuli [7]. TRPV1 is a polymodal ion channel activated by various stimuli (physical and chemical) such as heat, low pH, capsaicin, voltage, and mechanical force [11]. A number of animal toxins, including RhTx, BmP01, and DkTx evoke a prickling and burning sensation by targeting sensory nerve endings of the TRPV1 channel [12,13,14]. These aspects imply that the TRPV1 channel is an extremely grounded target for the defensive peptide toxins to produce pain by poisonous animals’ envenomation.

Although a collection of diverse proteins have been identified in caterpillar venom [15,16,17], the targets of these chemical weapons are poorly understood. In this study, we first collected crude venom from the *L. consocia* caterpillars and probed its bioactivities in a mouse model. Complementary animal behavior tests and electrophysiological recordings were used to unravel the molecular mechanism and biochemical tactics of the *L. consocia* venom-induced burning pain. Furthermore, we conducted a transcriptome analysis of the bristles and the epithelium at the base of the cortical region. These results together form the foundation for a better understanding of the strategy behind pain induction by the *L. consocia* venom.

## 2. Results

### 2.1. Caterpillar Venom Induces Pain in a Mouse Model In Vivo

Venom was extracted from the *L. consocia* caterpillar bristles, and its pain producing effect was assessed. An in vivo injection of 10 μL of the bristles extract (100 μg/mL, refer to Experimental Section 5.2) to the hind paw of WT mice led to long-lasting paw licking responses (Figure 1B). This observation suggests that a chemical rather than physical mechanism underlies the pain-producing ability of caterpillars. We fractioned the bristles extract by using a C_4_ RP-HPLC (Waters, Milford, CT, USA) column; a typical chromatogram of the venom is shown in Figure 1C. After separation, all the protein fractions were collected according to the absorbance at 280 nm, and all the eluted fractions were combined as “crude venom” (refer to Experimental Section 5.3) of *L. consocia.* The bristles extract and the crude venom were compared for pain-producing abilities, using the paw-licking duration of a mouse model. We found that both the bristles extract and crude venom exhibited strong pain-producing activities (Figure 1D), which suggests that the protein component of the *L. consocia* venom was the ‘biochemical weapon’ to induce intense pain symptom. Since the bristles extract and crude venom elicited similar pain response, the crude venom was used for further assays.

### 2.2. Caterpillar Venom Targets TRPV1 Ion Channel

The *L. consocia* crude venom could elicit intense pain responses after subcutaneous injection into the WT mice. To identify the potential target, we screened crude venom on several pain-related ion channels, which included the dorsal root ganglion sodium channel (DRG-Na), dorsal root ganglion potassium channel (DRG-K), dorsal root ganglion calcium channel (DRG-Ca), acid-sensing ion channel 2a (ASIC 2a), P2X ligand-gated ion channel 3 (P2X3), KCNQ4, transient receptor-potential M8 (TRPM8), TRPV2, and TRPV1 channels. For this experiment, 100 μg/mL of *L. consocia* crude venom was used for the functional screening of these channels and the response of each channel type was determined using whole-cell patch clamp recordings (Figure 2). Among these ion channels, we found that 100 μg/mL of *L. consocia* crude venom evoked substantial currents from TRPV1 (Figure 2I) but not other channels in acute dissociated mice DRG neurons (Figure 2A–C) and transiently transfected human embryonic kidney 293T (HEK293T) cells (Figure 2D–H), therefore suggesting that the *L. consocia* crude venom specifically and potently activates the TRPV1 channel. As an indication of strong TRPV1 activation, the venom-induced current was similar in amplitude to that of the capsaicin-induced current. The results indicate that TRPV1 is an important target in the *L. consocia* caterpillars’ induction of a painful sensation.

### 2.3. Calcium Signal from Sensory Neurons Challenged with Caterpillar Venom

As the *L. consocia* crude venom potently activates the mouse TRPV1 channel, and TRPV1 channel is primarily expressed in mouse DRG neurons [18], we carried out calcium imaging of mice DRG neurons challenged with the *L. consocia* crude venom. As shown in Figure 3A, we found that the crude venom elicited intracellular calcium signals in all capsaicin-responsive neurons that were as robust as capsaicin, but not in capsaicin-irresponsive neurons. As expected, when tested in DRG neurons of the TRPV1 KO mice, no notable changes were observed (Figure 3B,C). These observations further indicate that the *L. consocia* crude venom directly activates the TRPV1 channel. 

### 2.4. L. consocia Crude Venom Induces Pain Behaviors in WT but Not TRPV1 KO Mice

To further evaluate if TRPV1 is a target of the pain-inducing toxins from the *L. consocia* caterpillars in vivo target, the crude venom effect was tested in both groups of WT and TRPV1 KO mice. Upon injection of 100 μg/mL of crude venom into the hind paw of the WT and TRPV1 KO mice, a remarkable pain behavior was observed for the WT mice unlike that of the TRPV1 KO mice (Figure 4A,B). To test whether the *L. consocia* crude venom produces pain via the TRPV1 pathway, capsazepine, a TRPV1 specific antagonist, was used to neutralize the effects of capsaicin. Ten microliters of crude venom (100 μg/mL) along with capsaicin (500 μM), capsazepine (2 mM), capsaicin/ capsazepine mixture, and crude venom/capsazepine mixture were used to evaluate the pain-induced behavior in both WT and TRPV1 KO mice, and an equal volume of saline was used as the control. The durations of biting/licking are summarized in Figure 4A,B, which showed that *L. consocia* crude venom and capsaicin induced pain in WT mice (Figure 4B), whereas both were substantially attenuated in the TRPV1 KO mice (Figure 4B). Moreover, the TRPV1 channel inhibitor, capsazepine, weakened the pain-inducing effects of both the crude venom and capsaicin (Figure 4A). These observations from the mice pain behavior assay, in combination with the electrophysiological and calcium-imaging tests on exogenously expressed TRPV1, further confirms that TRPV1 is in vivo target of the *L. consocia* crude venom.

### 2.5. Transcriptome Analysis of the Caterpillar L. consocia Venom Glands

To better understand the composition of the crude venom, a transcriptome analysis of the caterpillar *L. consocia* was conducted. For this purpose, a full-length cDNA library was constructed from the venom gland-like region of *L. consocia.* After sequencing, 115,515,164 clean reads were obtained for further bioinformatic analysis and functional annotation. Using the Trinity software (v2.4.0, GitHub, Inc.) to splice and cluster the short reads, a total of 126,670 universal genes (Unigene) were identified (Figure 4C). More than 70% of unigene sequences were less than 500 bp in length. For functional annotation, all unigene sequences were searched for NCBI non-redundant protein database (Nr) using the Blastx tool, with a cut-off E-value of 10^−5^. Matched unique sequences were aligned against the Gene Ontology (GO); the best hits were used to annotate the sequences (Figure 4D). According to sequence annotation, a total of 7368 unigenes were assigned to GO terms, including 3,733 with hits at the biological process level, 1855 at the cellular component level, and 6,322 at the molecular function level. In addition, a total of 162 defense mechanism-related genes were highlighted according to the annotation (Appendix A). The caterpillar *L. consocia* venom gland-like tissue transcriptome analysis revealed extreme structural and functional diversities of the venom components, which indicates that the *L. consocia* venom may adopt multiple defense strategies for coordinated defense. Our transcriptome results will assist the identification of functional molecules that contribute to the physiology of caterpillars.

## 3. Discussion

Caterpillars are widely distributed across the globe. Their venoms comprise of multifunctional peptide toxins as a kind of high-efficiency defense equipment [5,19]. Envenomation by caterpillars is known to produce pain, however, the mechanism involved in pain induction is still elusive. This work presents the first insight into the biological activity of caterpillar venom. Both genetic and pharmacological assays indicate that TRPV1 serves as one of the primary nociceptors in caterpillar-induced pain. Unlike the venom of hymenopterans [19,20], which also elicits pain to deter potential predators, we failed to identify pore-forming toxins in the venom of *L. consocia*. This observation suggests that divergent evolution occurred in venomous insects, which makes venoms from insects vast and the main unexplored source of new animal toxins.

As an efficient defensive strategy, toxin-induced TRPV1 activation has been identified in several venomous animals [12,13,14,21,22]. The Chinese bird spider (*Ornithoctonus huwena*) exploits its bivalent toxin, DkTx, which results in the formation of a stable complex with the TRPV1’s outer pore that leads to relentless channel activation and severe pain behavior [14]. The Chinese red-headed centipede (*Scolopendra subspinipes)* has the capacity to cause extreme and localized pain immediately after centipede envenomation. It was found that centipede venom could reduce the heat activation threshold of the TRPV1 channel and activate TRPV1 at mammals’ physiological body temperatures [12]. In addition, scorpion venom could also target the TRPV1 channel to elicit intense pain in the mildly acidic (pH 6.5) venom environment [21]. Caterpillar venoms are a well-known pain-producing chemical punch. However, caterpillar venoms have not been as well studied as the venoms of spiders, centipedes, and scorpions [23,24,25,26,27,28,29]. Our present study reveals that *L. consocia* crude venom elicits intense pain behaviors by targeting the TRPV1 channel. Given the extensive cost to synthesize venom, poisonous animals frugally use venom to disrupt essential targets. The TRPV1 channel is widely distributed in peripheral nociceptive neurons and a good target to induce pain. Accordingly, venomous animals often evolve toxins to target the TRPV1 channel, resulting in severe pain behaviors. Together with previous studies of the TRPV1-targeting toxins from spiders, scorpions, and centipedes, our present study of the caterpillar venom suggests the existence of convergent evolution in pain-producing toxins.

Besides TRPV1, *L. consocia* crude venom probably targets multiple pain-related receptors. As shown in Figure 4A,B, the TRPV1 channel specific inhibitor, capsazepine, could not completely eliminate the mice paw licking behavior elicited upon crude venom injection. Moreover, the crude venom elicited a minimal pain symptom in TRPV1 KO mice compared with WT mice. These results indicate that the TRPV1 channel is not the only nociceptor of *L. consocia* crude venom, and there are possibilities of having other targets, which synergistically with TRPV1 elicit severe painful sensations. Moreover, we can’t rule out the indirect role of TRPV1 in caterpillar venom-induced pain. For instance, inflammatory molecules such as TNF-α [30] and IL-1β [31] may participate in the TRPV1-related pain sensation. Therefore, the research on the pain receptor related to the envenomation of caterpillar venom needs to be further deepened. In our study, the total protein of *L. consocia* venom is the main component accounting for intense pain, and the key molecules in crude venom also need to be further excavated.

Similar to other venomous animals, *L. consocia* produces crude venom that exhibits intense diversity in amino acid sequences. Biochemical studies on the composition of the caterpillar venom have been hindered by the scarcity of the venom distributed among the bristles. Taking advantage of advanced molecular approaches, our study has analyzed the biochemical nature of the venoms. A total of 126,670 universal genes (UniGene) were assembled and the median length of the UniGene was 321 bp. The results demonstrate that *L. consocia* caterpillar venom is rich in short peptide fragments. Moreover, the discovery of 162 defense mechanism-related genes may help us to better understand the defense tactics of caterpillars, though much more pain-related conserved genes need to be further explored. Given that animal toxins are good resources for pharmacological research and clinic therapeutics, results from our study provide the opportunity to seek leading drugs and probes.

## 4. Conclusions

To conclude, our present study provides a biological and biomedical depiction of the crude venom of *L. consocia.* The properties of the venom reported here show that it possesses components which elicit severe pain behaviors in animal models. Furthermore, pain-related receptor screening identified TRPV1 as a candidate target of the *L. consocia* venom. Compared with TRPV1 KO mice DRG neurons, there is a substantial boost in intracellular calcium concentrations on wild-type mice DRG neurons upon venom treatment. In addition, the TRPV1 channel antagonist potently attenuated both crude venom and capsaicin induced pain behaviors. Together our results suggest that *L. consocia* crude venom potently activates the TRPV1 channel, thus causing severe pain. Transcriptome analysis provided opportunities to further investigate these venom components. Our study may lead to interesting new perspectives for the defense strategy of the *L. consocia* caterpillar. Future studies will identify unknown components which could exhibit novel pharmacological properties. 

## 5. Experimental Section

### 5.1. Animals

C57BL/6J Mice (aged 6–8 week, 23–28g, both sexes) were obtained from the Laboratory Animal Research Center of Kunming Medical University, China. TRPV1-KO mice (aged 6–8 week, 23–28g, both sexes) were obtained from the Model Animal Research Center of Nanjing University, China. All the mice were bred in the laboratory animal center at Kunming Institute of Zoology, Chinese Academy of Sciences (CAS). All animal experimental protocols were approved by the Institutional Animal Care and Use Committees at Kunming Institute of Zoology, CAS (identification code: KIZ-SMKX-2018091001; date of approval: 10 September 2018). Efforts were made to minimize the number of animals used and unnecessary suffering.

### 5.2. Venom Collection

The caterpillars (*Latoia consocia*) (both sexes, *n* = 1800) were captured from rural areas of Yong Ping, Yunnan Province, China. The live caterpillars were transported and maintained in the insectarium of Kunming Animal Museum. The bristles extracts were collected by manually removing the bristles from the back of the larvae, and homogenized in PBS (0.1M, pH = 7.4) solution containing 10 μL proteinase inhibitor (P8340-5, Sigma-Aldrich Co., Ltd. Shanghai, China) by a tissue grinder (Qiagen, Courtaboeuf, France). The mixture was filtrated with a strainer mesh (200 mesh, 74 μΜ in diameter, Solarbio Co.,Ltd. Beijing, China), and the effusion was centrifuged at 12,000 rpm, 10 min, 4 °C. The supernatant was collected and lyophilized, and the dry extract was named “bristles extract” of *L. consocia*. The bristles extract was used to check for pain behavior in the mouse model.

### 5.3. Venom Purification

0.5 mL bristles extract diluted with 2.0 mL 0.1 M PBS (pH = 7.4) was filtered and centrifuged at 12000 rpm at 4 °C for 10 min. The supernatant was loaded and purified using RP-HPLC on a C4 column (XBridge protein BEH C4 OBD TM Prep Column, 300 Å, 5 μm, 10 mm × 250 mm, 1/pkg). Elution was carried out with a linear gradient of solution B (Acetonitrile with 0.1% TFA) at a flow rate of 1.5 mL/min and the absorbance of the elution fractions was monitored at 280 nm. The eluted peptides were collected and lyophilized, and all the eluted fractions were combined as “crude venom” of caterpillar *L. consocia*. The crude venom was weighed and dissolved in saline or a different electrophysiological external solution for further use.

### 5.4. Dorsal Root Ganglion Isolation

DRG neuron acute separation solutions included Oxygen-enriched Dulbecco’s modified Eagle’s medium (DMEM) solution (346 mg DMEM powder and 74 mg NaHCO_3_ was dissolved in 20 mL of ultrapure water and charged with medical oxygen for 10 min) and digestive solution. After decapitating C57BL/6J mice, the spines were separated and placed in oxygen-rich DMEM (Invitrogen, Carlsbad, CA, USA) solution for 5 min, and the DRG neurons were picked out using the venus scissors leaving only the transparent spherical ganglion. The transparent ganglion was placed in an oxygen-rich DMEM solution, and completely cut with venus scissors before digestion. The manually cut ganglion was placed in 5 mL digestive solution (1 mg trypsin and 2.2 mg collagenase were dissolved in 5 mL of oxygen-rich DMEM solution) and digested in a 37 °C shake bed at 100 rpm about 15–30 min. After digestion, 1.2 mg trypsin inhibitor (Sigma-Aldrich Co.,Ltd. Shanghai, China) was added into the 5 mL digestive solution with ganglion to stop digestion process. Then the digestive solution with ganglion was centrifuged at 1000 rpm, 37 °C for 5 min. After removal of the liquid, DRG neurons were cultured in Dulbecco’s modified Eagle’s medium (DMEM) with 10% fetal bovine serum (FBS) and 1% penicillin/streptomycin, incubated at 37 °C with 5% CO_2_ for further use.

### 5.5. Cell Preparation and Transient Transfection

HEK293T cells were cultured in cell incubator with DMEM medium, incubated at 37 °C with 5% CO_2_. Cells were transfected with TRPV1, mASIC2a, mKCNQ4, hP2X3, mTRPV2, and mTRPM8 cDNA and enhanced with green fluorescent protein (eEGFP) plasmid by lipofectamine 3000 and P 3000 (Thermo Fisher Scientific, Waltham, MA, USA) following the manufacturer’s protocol. The cells were later digested with 0.25% trypsin between one and two days after transfection. Whole cell patch clamp recordings were conducted after the cells were attached to the glass slide.

### 5.6. Calcium Fluorescence Imaging

Mice DRG neurons were treated with Fluo-4 AM in Ringer’s solution—5 mM KCl, 140 mM NaCl, 2 mM MgCl_2_, 10 mM D-glucose, 10 mM HEPES, and 2 mM CaCl_2_, pH 7.4 adjusted with NaOH. Olympus IX81 microscope and Hamamatsu C4742 charge-coupled device (CCD) camera controlled by the MetaFluor Software (Molecular Devices, Sunnyvale, CA, USA, 2016) were used to obtain fluorescence images of the DRG neurons. Fluo-4 was excited using a mercury vapor light source with a 500/20-nm excitation filter, while the fluorescence emission was monitored with a 535/30-nm filter. The fluorescence image needs to be analyzed with an automatic program written by MetaMorph software (Molecular Devices, Sunnyvale, CA, USA, 2016) and Igor Pro (v6.1.2, WaveMetrics, Inc., Lake Oswego, OR, USA, 2015).

### 5.7. Electrophysiological Recordings

Electrophysiological recordings were performed at room temperature using an EPC-10 amplifier (HEKA Electronik, Lambrecht, Germany) as stated in the previous report [32]. Patch pipettes were pulled from the borosilicate glass capillary tube, and when filled with internal solution, their resistances were 3–5 MΩ. For KCNQ channels recording, the internal solution contained 160 mM KCl, 5 mM MgCl_2_, and 5 mM HEPES (pH 7.2); and external solution contained 160 mM NaCl, 2.5 mM KCl, 1 mM MgCl_2_, 2 mM CaCl_2_, 10 mM HEPES, and 10 mM D-glucose (pH 7.4). For TRP channels recording, the internal solution and external solution contained 3 mM HEPES, 0.2 mM Ethylene Glycol Tetraacetic Acid, pH 7.2, and 130 mM NaCl. For voltage-gated sodium channels, the internal solution contained 5 mM HEPES, 10 mM NaCl, and 135 mM CsF at pH 7.0; and the external solution contained 5 mM CsCl, 30 mM NaCl, 1 mM MgCl_2_, 25 mM D-glucose, 1.8 mM CaCl_2_, 20 mM triethanolamine-chlorine, 70 mM tetramethylammonium chloride, and 5 mM HEPES at pH 7.4. For voltage-gated calcium channels, the external recording solution contained 0.0025 mM tetrodotoxin, 25 mM Tetraethylammonium Chloride (TEA-Cl), 130 mM choline chloride, 3 mM KCl, 0.6 mM MgCl_2_, 5 mM BaCl_2_, 10 mM HEPES, 4 mM glucose, 1 mM NaHCO_3_, and pH 7.4 adjusted with NaOH; and the internal solution contained 140 mM CsCl, 2 mM MgCl_2_, 0.1 mM CaCl_2_, 2 mM ATP, 10 mM HEPES, and pH 7.2 adjusted with Tris. For voltage-gated potassium channels, the pipette solution contained 10 mM EGTA, 140 mM KCl, 4 mM Mg-ATP, and 10 mM HEPES, pH 7.2 adjusted with KOH; and the bath solution contained 5.4 mM KCl, 135 mM NaCl, 0.33 mM NaH_2_PO_4_, 1 mM MgCl_2_, and 10 mM HEPES, 10 mM Glucose, pH 7.4 adjusted with NaOH. For ASIC channel recording, the internal solution contained 10 mM HEPES, 120 mM KCl, 0.5 mM CaCl_2_, 1 mM MgCl_2_, 2 mM Mg^2+^-ATP, 5 mM EGTA, and 30 mM NaCl, pH 7.25, adjusted with Tris Base; and the bath solution contained 5 mM KCl, 150 mM NaCl, 2 mM CaCl_2_, 1 mM MgCl_2_, 10 mM HEPES, and 10 mM glucose, pH 7.4, adjusted with NaOH. For P2X channel recording, the internal solution contained 5 mM EGTA, 120 mM KCl, 30 mM NaCl, 0.5 mM CaCl_2_, 1 mM MgCl_2_, and pH 7.4; and the external solution contained 5 mM KCl, 150 mM NaCl, 10 mM HEPES, 10 mM glucose, 1 mM MgCl_2_, and 2 mM CaCl_2_. All biochemical reagents used above were purchased from sigma (Sigma-Aldrich Co.,Ltd. Shanghai, China).

### 5.8. Paw Licking Behavior Test

Upon adapting to the surrounding environment at 25 °C for 30 min, mice were injected intraplantarally on the left hind paw with 10 µL of respective solutions with proper concentrations (capsaicin 500 μM, capsazepine 2 mM, bristles extract 100 μg/mL, and crude venom 100 μg/mL) at the plantar surface of the left hind paw (*n* = 3–6). The control group of mice was injected with an equal volume of saline. A digital camera was used to record the licking time for 30 min immediately after the injection.

### 5.9. Transcriptome Sequencing Analysis

A directional mRNA was extracted from the part of tissue resembling venom (bristles of the back of the larvae with skin tissue about 500 mg) using the Trizol Reagent extraction according to the previous report [33]. RNA from the *L. consocia* tissue was isolated using the RNeasy Mini Kit (Qiagen, Courtaboeuf, France) according to the manufacturer’s protocol. The total RNA were used for the construction of sequencing libraries which were then subjected to DSN normalization following the procedures laid out in the NEBNext Ultra RNA Library Prep Kit (New England Biolabs, Inc., Ipswich, MA, UK), and each library was sequenced on a hiseq X ten as stated in manufacturer’s guidelines, giving rise to 150-bp paired-end reads (NextOmics, Inc., Wuhan, China). The Trinity (V2.4.0) was used to splice the short sequence and the trinity sequence with the longest sequence was selected as the Unigene. After all the Unigenes have been proceeded GO annotation, WEGO software was used to do GO function statistic classification in order to understand the distribution characteristics of gene function.

### 5.10. Data Analysis

The data recorded by electrophysiology were obtained and analyzed by Pulse program and Igor Pro (v6.1.2, WaveMetrics, Inc., Lake Oswego, OR, USA, 2015). All data obtained from the animal experiments and transcriptome were analyzed using GraphPad Prism 7 (Version 7.00, GraphPad Software, Inc., San Diego, CA, USA, 2018). The data points of all experiments are shown as the means ± S.E.M. and n is presented as the number of animals or cells used for each set. Student’s *t* test was used for statistical analysis of the data, and significant difference (*p* < 0.05) was indicated by *.

### 5.11. Data Availability

Data supporting the results of this work are available within the paper and Appendix A. Sequencing data that support the transcriptome results of this study have been deposited in the NCBI Sequence Read Archive (SRA) and are accessible under the SRA accession PRJNA588657.

## Figures and Tables

**Figure 1 toxins-11-00695-f001:**
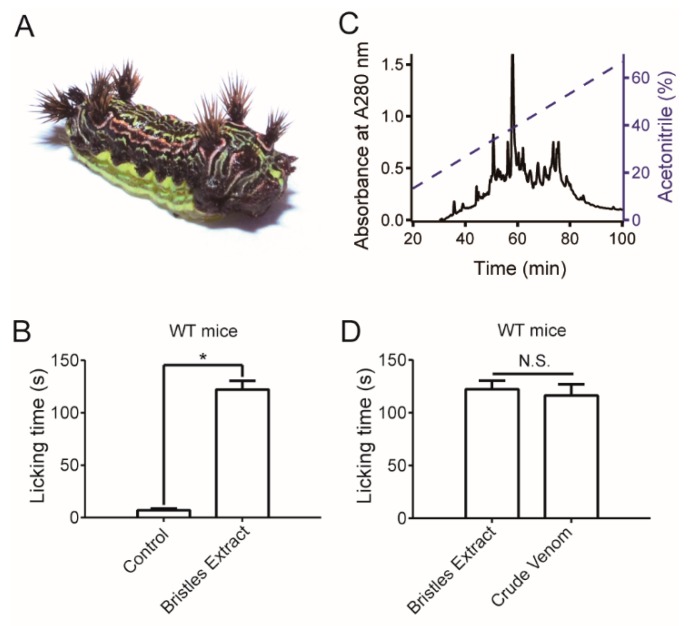
The *Latoia consocia* caterpillar venom induces pain in WT mice. (**A**) Representative image of the *Latoia consocia* caterpillar. (**B**) Ten microliters saline and bristles extract (100 μg/mL) injected into WT mice exhibited a remarkable pain behavior difference between saline and the bristles extract. Two-sided *t*-test: *, *p* < 0.05; *n* = 3. (**C**) Separation of the bristles extract of *L. consocia* on a C_4_ RP-HPLC column. (**D**) Both the bristles extract (100 μg/mL) and crude venom (100 μg/mL) injected into WT mice exhibited intense pain behavior, and there was no significant difference between them. Two-sided *t*-test: N.S., not significant; *n* = 3.

**Figure 2 toxins-11-00695-f002:**
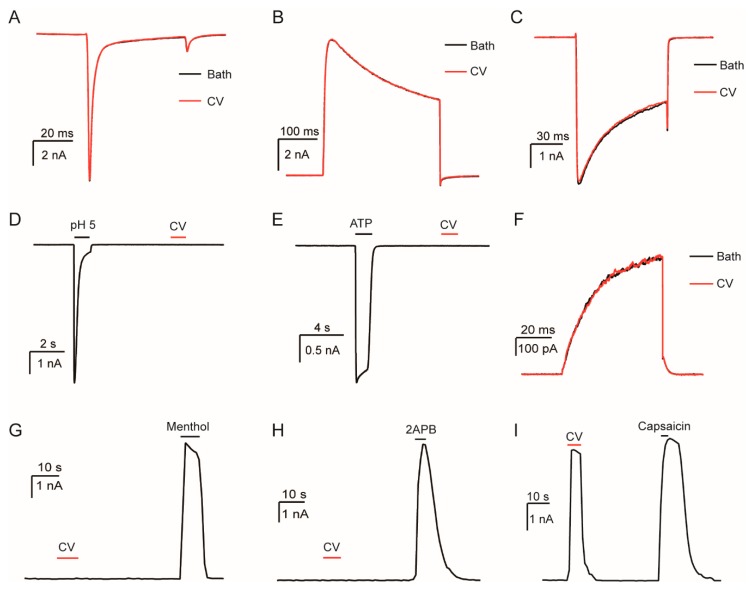
Selectivity of the *L. consocia* caterpillar crude venom on pain-related ion channels. The selectivity of 100 μg/mL *L. consocia* crude venom on mice dorsal root ganglion (DRG) sodium channel (**A**), potassium channel (**B**) and calcium channel (**C**); The selectivity of 100 μg/mL *L. consocia* crude venom on mASIC2a channel (**D**), hP2X3 channel (**E**), mKCNQ4 channel (**F**), mTRPM8 channel (**G**), mTRPV2 channel (**H**), and mTRPV1 channel (**I**).

**Figure 3 toxins-11-00695-f003:**
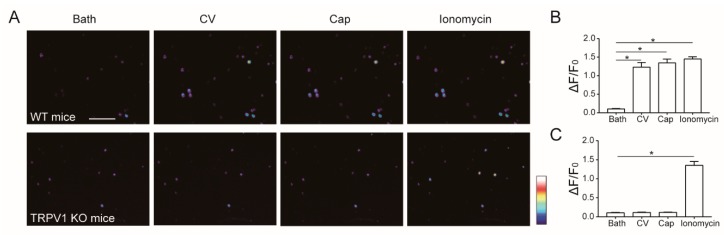
Calcium imaging of mice DRG neurons. (**A**) DRG neurons calcium imaging of WT (top row) or TRPV1 KO (bottom row) mice sequentially administered with *L. consocia* crude venom (100 μg/mL), capsaicin (10 μM), and ionomycin (1 mM). Scale bar, 250 mm. Representative calcium fluorescence signals of DRG neurons from WT (**B**) or TRPV1 KO mice (**C**). Two-sided *t*-test: *, *p* < 0.05; *n* = 3 cells.

**Figure 4 toxins-11-00695-f004:**
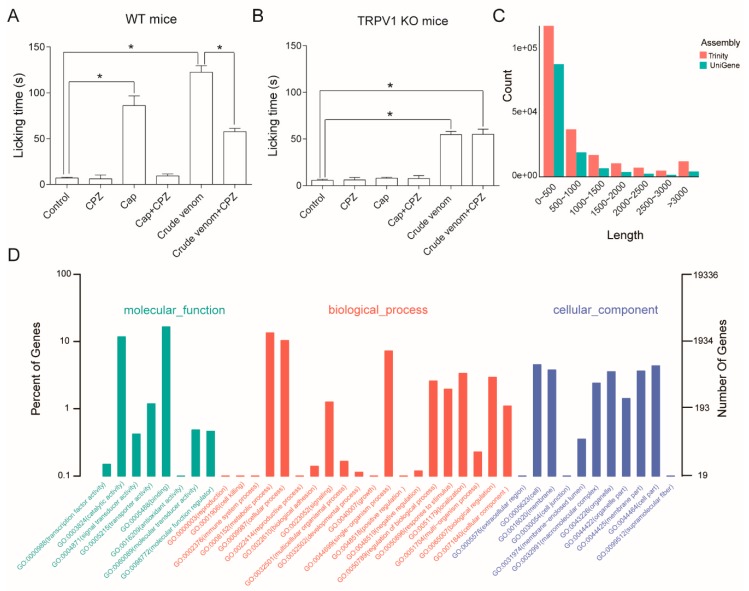
Mean duration of venom-induced paw licking and venom gland-like region transcriptome analysis. (**A**) Mean durations of paw licking induced by 10 μL of saline (control), capsazepine (CPZ, 2 mM), capsaicin (Cap, 500 μM), crude venom (100 μg/mL), capsaicin (500 μM)/capsazepine (2 mM) mixture, and crude venom (100 μg/mL)/capsazepine (2 mM) mixture injected into the left hind paw of WT mice. Two-sided *t*-test: *, *p* < 0.05; *n* = 6. (**B**) Mean durations of paw licking induced by 10 μL of saline (control), capsazepine (2 mM), capsaicin (500 μM), crude venom (100 μg/mL), capsaicin (500 μM)/capsazepine (2 mM) mixture, and crude venom (100 μg/mL)/capsazepine (2 mM) mixture injected into the paw of TRPV1 KO mice. Two-sided *t*-test: *, *p* < 0.05; *n* = 6. (**C**) Length distribution of assembly sequence and Unigene sequence of the *L. consocia* caterpillar venom gland-like region transcriptome. (**D**) Gene ontology (GO) analysis of the caterpillar *L. consocia* venom gland-like region transcriptome unigenes according to their involvement in the biological process, cellular component, and molecular function.

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
