# Peer review of "The Latoia consocia Caterpillar Induces Pain by Targeting Nociceptive Ion Channel TRPV1"

_toxins, 2019, doi:10.3390/toxins11120695_

Round 1
Reviewer 1 Report
The study aimed to evaluate the pain induced by caterpillar venom and to characterize the target of this effect. The work was well conducted and the results are important since the effect of caterpillar on pain is not characterized. However, some revision is necessary in the manuscript
- Standardize letter size and type along the introduction
- There are some English mistakes that should be revised. An English review is indicated
Experimental section
5.1. Animals – include age, weight and sex from the used mice
Line 210 – replace “was” by “were”
5.2. Bristles were manually homogenized? Venoms store were done without any kind of filtration or centrifugation?
5.4. Confirm how the ganglions were dispersed. Is there any step before digestive juice, like manual dissociation? Describe what does “juice” mean. What kind of enzyme was used for this digestion?
5.4. Why were used rats and mice for DRG cultures? Cells from which animal were used in each assay? Authors should describe it and discuss which differences were noted comparing cells from each one, and clearly show it in the results
5.8 Indicate the concentrations used for each assay
Fig. 1D – there is no significance indicated in the figures. In addition, how was performed the statistical analyses of these data, since in the item 5.10 it was described that it was used Student´s t test but here you have more than two different groups?
Fig 1D. How much of each fraction was injected in the animals? With how much protein?
Line 77 – “Among these ion channels, we found that 100 μg/ml L. consocia crude venom evoked substantial currents from TRPV1 but not other channels in transiently 79 transfected human embryonic kidney 293T (HEK293T) cells” – indicate in the text the figures for each result
Fig 2. – Figures 2 demonstrates the effect of CV and bath but not of fractions on currents from DRG cells, as described in line 71. Please, include the results of the fractions.2
2.4 – Although the results strongly suggest TRPV1 as the main target for caterpillar venom, in the figure 4C it can be seen that crude venom still induces relevant pain in TRPV1 KO mice. Comparing to figure 4A it can be seen that absence of TRPV1 reduces the pain in only 50%. It means that there are other targets for venom action. Authors could not ignore this fact and discuss it.
Why did the authors do not purify fractions F3 and F6? I do not believe that the results demonstrated in figure 1 C and 1 D contribute to the manuscript, since F 3 and F 6 were not purified, and since the results from these fraction were not included in the next assays. My suggestion is to exclude them from the paper and to publish it later, when authors get the identification of the components responsible for pain effect. But it is just a suggestion. The authors can decide about it or not.
Discussion can be improved, discussing TRPV1 as not the only possible target for this pain. In additon, after the decision about to keep or exclude the results from the fractions, it should also be better discussed
Author Response
The study aimed to evaluate the pain induced by caterpillar venom and to characterize the target of this effect. The work was well conducted and the results are important since the effect of caterpillar on pain is not characterized. However, some revision is necessary in the manuscript.
Response: We deeply appreciate the reviewer’s comment to the importance of this study. We have thoroughly revised the manuscript to address the concerns raised by the reviewer.
- Standardize letter size and type along the introduction
Response: Addressed.
- There are some English mistakes that should be revised. An English review is indicated
Response: Thank you. We have carefully corrected grammar errors throughout the revised manuscript.
Experimental section
5.1. Animals – include age, weight and sex from the used mice
Response: Fixed.
Line 210 – replace “was” by “were”
Response: Fixed.
5.2. Bristles were manually homogenized? Venoms store were done without any kind of filtration or centrifugation?
Response: We have added a detailed description of venom collection in the methods section of the revised manuscript. We have rephrased the venom extraction as below:
“Bristle extracts were collected by manually removing the bristles from the back of the larvae, and homogenized them in phosphate-buffered saline (PBS, 0.1 M, pH = 7.2) solution containing 10 μl proteinase inhibitor cocktail (Sigma, P8340-5, St. Louis, MO, USA) by tissue grinder (QIANGEN). The mixture was filtrated with strainer mesh (200 mesh, 74 μΜ in diameter, Solarbio), and the effusion was centrifuged at 12000 rpm for 5 min, 4°C. The supernatant was collected and lyophilized, and the dry extract was named “venom extract” of L. consocia. Venom extract was used to check pain behavior in mouse model”.
5.4. Confirm how the ganglions were dispersed. Is there any step before digestive juice, like manual dissociation? Describe what does “juice” mean. What kind of enzyme was used for this digestion?
Response: We have added a detailed description of dorsal root ganglion dispersal and manual dissociation procedure. We have replaced the word “juice” with “solution”, and the digestion enzymes were also clearly introduced.
5.4. Why were used rats and mice for DRG cultures? Cells from which animal were used in each assay? Authors should describe it and discuss which differences were noted comparing cells from each one, and clearly show it in the results
Response: Thank you for pointing it out. DRG neurons of both rats and mice express several pain-related ion channels, such as voltage-gated sodium channel, voltage-gated potassium channel, voltage-gated calcium channel and TRPV1 channel. In our study, only mice DRG neurons were used and the rats DRG neurons were mistakenly mentioned. We appreciate the reviewer’s suggestion, and we have added the origin (WT or TRPV1 KO mice) of cells used in each assay.
5.8 Indicate the concentrations used for each assay
Response: Indicated.
Fig. 1D – there is no significance indicated in the figures. In addition, how was performed the statistical analyses of these data, since in the item 5.10 it was described that it was used Student´s t test but here you have more than two different groups?
Response: Statistical analysis was carried out by Student’s t test, and all the statistical significance analyses were compared with the control group, or otherwise specifically marked. The original Fig. 1D has been deleted in the revised manuscript.
Fig 1D. How much of each fraction was injected in the animals? With how much protein?
Response: We appreciate reviewer’s comments. In response to the reviewer’s suggestion that all the fractions were not purified and also were not used in other assays, we have decided to keep aside the fractions results in the revised manuscript.
Line 77 – “Among these ion channels, we found that 100 μg/ml L. consocia crude venom evoked substantial currents from TRPV1 but not other channels in transiently transfected human embryonic kidney 293T (HEK293T) cells” – indicate in the text the figures for each result
Response: Thank you for pointing it out. Fixed.
Fig 2. – Figures 2 demonstrates the effect of CV and bath but not of fractions on currents from DRG cells, as described in line 71. Please, include the results of the fractions.
Response: We appreciate reviewer’s comments. As mentioned above, we have deleted the fractions results in Figure 1D and indicated a need for further study in Discussion.
2.4 – Although the results strongly suggest TRPV1 as the main target for caterpillar venom, in the figure 4B it can be seen that crude venom still induces relevant pain in TRPV1 KO mice. Comparing to figure 4A it can be seen that absence of TRPV1 reduces the pain in only 50%. It means that there are other targets for venom action. Authors could not ignore this fact and discuss it.
Response: Thank you for pointing it out. A brief discussion has been added to discuss the possibilities of having other targets for venom-induced pain sensation.
Why did the authors do not purify fractions F3 and F6? I do not believe that the results demonstrated in figure 1 C and 1 D contribute to the manuscript, since F 3 and F 6 were not purified, and since the results from these fraction were not included in the next assays. My suggestion is to exclude them from the paper and to publish it later, when authors get the identification of the components responsible for pain effect. But it is just a suggestion. The authors can decide about it or not.
Response: Encouraged by the comments of this reviewer, we have decided to keep aside the fractions results in Figure 1D. Figure 1C is a necessary procedure to obtain “crude venom”, which has been described in Experiment section 5.3.
Discussion can be improved, discussing TRPV1 as not the only possible target for this pain. In additon, after the decision about to keep or exclude the results from the fractions, it should also be better discussed
Response: We appreciate the reviewer’s suggestion. We have added a discussion on other possible targets for caterpillar venom-induced pain. A further need for fractions purification has been also highlighted in Discussion section.

Reviewer 2 Report
The manuscript intends to demonstrate that Latoia consocia caterpillar venom induces pain by targeting the TRPV1 ion channel. The major argument of the demonstration comes from the fact that TRPV1 KO mice do not show pain behavior.
The paper is interesting and well written. Not being a specialist of venoms, I have however several remarks.
1/ Is it possible that TRVP1 involvement, as demonstrated in KO mice, could reveal indirect? Presently, the authors just mention TRPV1 was mentioned in other studies related to e.g. Chinese bird spider venom. Naively, are there no more elements to provide/discuss about a proposed mechanism related to pain induction?
2/ This study focuses on Latoia consicia caterpillar. Could some information on related species comfort the present results? E.g. are there conserved genes that could be related to pain induction? If available, some elements in that direction could be of interest. Or at least, would it worth the question be set and the perspective discussed?
3/ The transcriptome analysis of the venom glands is welcome in the context of the study. However, the way it is presented sounds very preliminary, which lowers its impact. Are there convincing elements presented in this section? More in depth analysis seems required.
Author Response
The manuscript intends to demonstrate that Latoia consocia caterpillar venom induces pain by targeting the TRPV1 ion channel. The major argument of the demonstration comes from the fact that TRPV1 KO mice do not show pain behavior.
The paper is interesting and well written. Not being a specialist of venoms, I have however several remarks.
Response: We appreciate the reviewer’s comments. In the revised manuscript, we have carefully addressed the reviewer’s concerns.
1/ Is it possible that TRVP1 involvement, as demonstrated in KO mice, could reveal indirect? Presently, the authors just mention TRPV1 was mentioned in other studies related to e.g. Chinese bird spider venom. Naively, are there no more elements to provide/discuss about a proposed mechanism related to pain induction?
Response: We appreciate Reviewer’s insight. Our results suggest that TRPV1 serves as a primary nociceptor in caterpillar-induced pain. However, we can't rule out the indirect role of TRPV1 in caterpillar venom-induced pain. For instance, inflammatory molecules such as TNF-α (Malek N. et. al. Mol Cell Neurosci, 2015) and IL-1β (Kim M.J. et. al. Pain, 2014) may participate in TRPV1-related pain sensation. Moreover, there are possibilities of having other targets for caterpillar venom-induced pain. Both the indirect role of TRPV1 and the possibilities of other targets have been added in Discussion section.
2/ This study focuses on Latoia consicia caterpillar. Could some information on related species comfort the present results? E.g. are there conserved genes that could be related to pain induction? If available, some elements in that direction could be of interest. Or at least, would it worth the question be set and the perspective discussed?
Response: We appreciate Reviewer’s insight. At present, the caterpillar venoms have not been well studied, and pain-related conserved genes are not identified. This is part of the reason that we have conducted transcriptome analysis of the caterpillar L. consocia venom gland region. A total of 162 defense mechanism-related genes identified by this effort have been shown in Supplementary data. A brief discussion on defense genes has been added in Result and Discussion section.
3/ The transcriptome analysis of the venom glands is welcome in the context of the study. However, the way it is presented sounds very preliminary, which lowers its impact. Are there convincing elements presented in this section? More in depth analysis seems required.
Response: Thank you for pointing it out. We have shown defense mechanism-related genes in Supplementary section, and sequencing data that support the findings have been deposited in the NCBI Sequence Read Archive (SRA) and are accessible under the SRA accession PRJNA588657.

Reviewer 3 Report
This paper describes characterization of the pain producing properties of caterpillar venom and identifcation of TRPV1 as a molecular target. The paper is interesting and well written and I would recommend acceptance after a few minor points have been addressed:
In section 2.1, please specify what the concentration of “100 ug/mL” for the crude venom refers to ? Does this refer to a total protein concentration, extract dry weight, or bristle weight ? Please specify how this value was determined ?
In section 2.2 please specify if the activity seen is caused by crude venom or some of the HPLC fractions. For example in the text the authors state that crude venom and fractions were tested on ion channels, but the data shown in Fig. 2 only refer to crude venom (CV) ? Please specifically state if any of the fractions had activity in the ion channel assay, or if the activity was lost as a result of purification.
In figure 4, please spell out “Cap” . or define in the legend.
In section 2.4 the pain-causing effects of crude venom are not entirely eliminated in the TRPV1 KO mice. This suggests that TRPV1 is not the only target and the authors perhaps should discuss this in more detail ?
In section 2.5. A more detailed analysis of the transcriptome data would greatly strengthen this paper. As it stands this section doesn’t add anything significant to the story of this paper as most of the GO terms refer to basic housekeeping genes that do not allow any conclusions to be drawn about specific venom components. In any case, the raw reads (data) from the RNAseq experiment should be deposited in a public database e.g the Sequence Read Archive (SRA) at the NCBI. Please provide an accession number.
Author Response
This paper describes characterization of the pain producing properties of caterpillar venom and identifcation of TRPV1 as a molecular target. The paper is interesting and well written and I would recommend acceptance after a few minor points have been addressed:
Response: We thank the reviewer for the encouraging comments.
In section 2.1, please specify what the concentration of “100 ug/mL” for the crude venom refers to? Does this refer to a total protein concentration, extract dry weight, or bristle weight? Please specify how this value was determined?
Response: Thank you for pointing it out. We have defined “crude venom” in Results section 2.1 and Experimental section 5.3. Briefly, after RP-HPLC purification, all the eluted fractions were collected and lyophilized, and combined together as “crude venom”. The crude venom was weighed and dissolved in saline or different electrophysiological external solution for respective experiments.
In section 2.2 please specify if the activity seen is caused by crude venom or some of the HPLC fractions. For example in the text the authors state that crude venom and fractions were tested on ion channels, but the data shown in Fig. 2 only refer to crude venom (CV) ? Please specifically state if any of the fractions had activity in the ion channel assay, or if the activity was lost as a result of purification.
Response: We appreciate the reviewer’s suggestion. Since venom fractions were not purified and also were not used in other assays, we have decided to keep aside the fractions results. In addition, we have added a brief discussion on need to for further purification of these fractions.
In figure 4, please spell out “Cap” . or define in the legend.
Response: Addressed.
In section 2.4 the pain-causing effects of crude venom are not entirely eliminated in the TRPV1 KO mice. This suggests that TRPV1 is not the only target and the authors perhaps should discuss this in more detail?
Response: We appreciate reviewer’s careful comments. A brief discussion has been added to discuss the possibility of having other targets of venom-induced pain sensation.
In section 2.5. A more detailed analysis of the transcriptome data would greatly strengthen this paper. As it stands this section doesn’t add anything significant to the story of this paper as most of the GO terms refer to basic housekeeping genes that do not allow any conclusions to be drawn about specific venom components. In any case, the raw reads (data) from the RNAseq experiment should be deposited in a public database e.g the Sequence Read Archive (SRA) at the NCBI. Please provide an accession number.
Response: Thank you for pointing it out. We have marked defense mechanism related genes in Supplementary section, and sequencing data that support the findings have been deposited in the NCBI Sequence Read Archive (SRA) and are accessible under the SRA accession PRJNA588657.

Round 2
Reviewer 1 Report
All of my concerns were adressed by the authors
Reviewer 2 Report
The authors have addressed adequately the remarks.